# Empirical Study on Human Movement Classification Using Insole Footwear Sensor System and Machine Learning

**DOI:** 10.3390/s22072743

**Published:** 2022-04-02

**Authors:** Wolfe Anderson, Zachary Choffin, Nathan Jeong, Michael Callihan, Seongcheol Jeong, Edward Sazonov

**Affiliations:** 1Department of Electrical and Computer Engineering, The University of Alabama, Tuscaloosa, AL 35487, USA; jwanderson4@crimson.ua.edu (W.A.); zmchoffin@crimson.ua.edu (Z.C.); esazonov@eng.ua.edu (E.S.); 2College of Nursing, The University of Alabama, Tuscaloosa, AL 35487, USA; mlcallihan@ua.edu; 3Department of Electrical Engineering, Pohang University of Science and Technology, Pohang 37673, Korea; somunza@postech.ac.kr

**Keywords:** movement classification, machine learning, smart shoe, footwear sensor, human movement classification

## Abstract

This paper presents a plantar pressure sensor system (P2S2) integrated in the insoles of shoes to detect thirteen commonly used human movements including walking, stooping left and right, pulling a cart backward, squatting, descending, ascending stairs, running, and falling (front, back, right, left). Six force sensitive resistors (FSR) sensors were positioned on critical pressure points on the insoles to capture the electrical signature of pressure change in the various movements. A total of 34 adult participants were tested with the P2S2. The pressure data were collected and processed using a Principal Component Analysis (PCA) for input to the multiple machine learning (ML) algorithms, including k-NN, neural network and Support-Vector Machine (SVM) algorithms. The ML models were trained using four-fold cross-validation. Each fold kept subject data independent from other folds. The model proved effective with an accuracy of 86%, showing a promising result in predicting human movements using the P2S2 integrated in shoes.

## 1. Introduction

Physical activity recognition is quickly becoming one of the most important methods of tracking human health and wellbeing. The desire to know more about our bodies has never been stronger with the rise of wearable devices such as smartwatches and smartphones. The rapid advances in the detection capabilities of these devices have shown just how far the technology can go, and they can detect human activity and movement with reasonable to high accuracy [1,2,3]. The demand for a device that can detect every type of daily human movement for the purpose of health tracking, injury prevention, and fall detection still exists. The need persists for a low-cost and non-invasive device. 

Many methods have been adopted to capture human movement. Among them, one of the most popular sensors used is inertial measurement units (IMU) [4,5,6,7]. Despite its high accuracy, the IMU revealed discomfort in daily use because of the amount of IMUs that need be attached to a body and the complex setup procedure. Video capture using cameras has proven to be relatively accurate as well [8,9,10,11]. However, camera-based motion sensing technology is not feasible when a direct line of sight obstructed. [11]. Other methods such as a millimeter wave [12] also offer good accuracy but pose the same problems for integration into real life. Acoustic signal [13] and infrared signal [14] have also been used. 

Another solution to this problem is to use a smart shoe using pressure-sensing technology in combination with machine learning. Smart shoe sensors present simple implementation to the body while providing comfort while in use. The smart shoe allows for practical use in day-to-day life. In addition, compared to other methods, this allows for a low-cost solution. The main areas of investigation in this field have been in pressure sensing materials and in the classification of different types of human movement. The use of multiple different materials such as sponges, textiles, and rubber for pressure sensing [15,16,17,18] shows promise for low-profile integration into an insole in a shoe. Pressure sensing for the use of human movement detection has also been investigated for cases such as stride counting [19], gait analysis [20], loss of balance and fall detection [21,22], and a variety of other human movements [23,24,25,26,27]. Thirteen frequently used household movements including lying, sitting, standing, walking, descend and ascend stairs, ergometer cycling, vacuuming, shelving items, washing dishes, sweeping the floor, and driving a car were classified in [23]. Walking and stair ascent and descent were classified in [28]. Walking up and down stairs were examined and classified in [29]. Two studies also investigated the detection of falling [21,22]. Based on previous research, there is a gap in knowledge related to more diverse movements, and the development of a system which can detect a broader set of human movements is in high demand. This will greatly improve the ability for workers to properly move and provide an approximation of estimated calories burned. Displayed in Table 1 is an overview of different study approaches and their accuracy in movement classification.

There are a number of footwear systems with the purpose of detecting the plantar pressure that are used today. One solution presented is an insole equipped with capacitive sensors with commercial solutions made by Moticon located in Munich, Germany [31]. Furthemore, others have been created for research purposes [32,33] in multiple different studies for gait tracking and motion analysis [21,34,35,36]. Other solutions such as the pedar© system designed by Novel in St. Paul, MN, USA [37] contain more than one hundred sensors so as to detect the precise pressure distribution across the foot [38,39]. A common solution was created with force-sensitive resistors placed at specific locations across the foot. This has been effectively used to detect pressure for a much lower cost than commercial solutions and allows for significant customizability [40,41,42,43,44,45].

This paper aims to detect thirteen different human movements using the P2S2 and machine learning algorithms. The P2S2 was developed in our previous work in [46] and the details will be provided in Section 2. It allows the system to acquire a more complete view of the user and makes it highly useful for injury prevention and health tracking. 

## 2. Materials and Methods

### 2.1. System Design

The basic concept of the pressure-sensing system relies on the use of force-sensitive resistor (FSR)s. These are sensors that are constructed with a substrate layer, a conductive film, a spacer, and another substrate with a conductive print on top. When a person’s foot pushes against the ground, a force is exerted back onto it through the shoe, which is known as ground reaction force (GRF). The GRF varies in magnitude and location depending the point of pressure on the foot while in active motion. When this force is applied to an FSR, the conductive film meets the conductive print on the bottom substrate. This contact increases with force. As this contact increases, the resistance decreases and more current flows. This resistance changes logarithmically for a linear increase in force. Because of this property, the amount of force could be measured.

Figure 1 shows the block diagram of the proposed human-movement sensor system that was used in our previous work [27,46]. The FSR sensors (Flexiforce A301) [47] were located at six common points of pressure across the foot. This included the inside (S1) and outside (S2) of the heel, the inside (S3), middle (S4), and outside (S5) of the midfoot, and under the big toe (S6). These sensors were connected to a microcontroller with a Bluetooth Low Energy module (Adafruit Feather M0 Bluefruit LE), a microSD card reader for data recording [48], and a 3.7 V lithium-ion battery. Each sensor was connected to an ADC terminal on the microcontroller to detect the analog signal from the pressure sensors. Drop-down resistors were connected between the ADC terminals and ground, and were then used to provide a threshold voltage for the pressure sensors. The sensors were placed on a flexible plastic substrate and copper strips were used to create a common power line and to route the six signals to the ADC terminals. Another layer of flexible plastic was placed over the sensors and copper strips to protect them from damage.

So as to provide for the largest possible participant pool, we found that the average shoe size in the United States for a male was 10.5 and for a female was 8.5, respectively [49]. For this reason, the pressure-sensing system was built into two different pairs of shoes: one a size 10.5 and one a size 8.5. The insole sensor system was placed underneath the included insole in each of the shoes, while the microcontroller was attached to the outside of each shoe with Velcro. A slit was made in the side of each of the shoes to feed the wiring from the insole sensor system to the microcontroller. Both shoes with their respective sensor systems are shown in Figure 2.

### 2.2. Movement Description

Data were acquired for thirteen different movements during testing. These movements were chosen as an extensive collection of movements that every person performs across the average day. The chosen movements are displayed below in Table 2.

### 2.3. Experimental Procedure

Testing for this study took place at the University of Alabama College of Nursing. Participants were provided with the shoes equipped with the insole pressure sensing system and were instructed to perform the series of movements. For start and end time verification, the participants were instructed to perform a heel raise before a new motion was performed. This is because the GRF profiles during a heel raise are prominent and easily distinguishable in consecutive movements. This was particularly important for movements such as squatting and stooping as to divide a test of multiple squats or stoops into individual movements for input to the machine learning algorithm.

A total of 34 subjects were tested for the study. This consisted of 12 males and 22 females with an average age of 22.6 years. All subjects provided written, informed consent to the study before any data were taken. Displayed in Table 3 is the information collected about each of the 34 test subjects.

### 2.4. Data Collection

Pressure data from the participants were collected from the P2S2 by using the microSD card reader to write to a text file. The written data included pressure data for each of the sensors, as well as a timestamp that was recorded in terms of the number of samples taken since microcontroller system start-up. The data were captured at a sampling rate of 50 Hz, corresponding to data capture every 20 ms ± 2 ms. This implies that the time needed to take one sample is 20 ms. The pressure data were recorded using a 10-bit ADC with the received values ranging from 0 to 1024. These were then scaled on a relative pressure scale of 0 to 100 for each sensor. Once the raw data were gathered, the text file generated on the SD card was imported onto a computer. Individual movement tests were separated from the text file using a MATLAB script which detected heel raises and separated them into individual movements. This script also normalized the time for all the samples. These individual tests were then visualized and processed into smaller pieces using the MATLAB Signal Analyzer. The movements of walking, running, stair ascent and descent, and pushing and pulling a cart were broken into folds of two steps each that overlap by one step each. For example, a test of ten steps of walking was split into nine individual datapoints, each of which consisted of two steps. For all other motions, each individual movement was captured. For example, a test where the subject squatted ten times was split into ten datapoints of one squat each. This trimmed data was then prepared for feature extraction and then input into the k-NN algorithm for training.

### 2.5. Machine Learning Technique

After the raw data were pre-processed, a MATLAB script was used to extract features from each data segment and normalize the sample numbering so that it would start at time zero. These features were as follows:Average value of each sensor (Feature 1 to 6). The average value of relative pressure was distinctive because each motion had a different period of pressure values. Each motion had different pressure values over the duration of the movement.Standard deviation of each sensor (Feature 7–12). The standard deviation feature was utilized for similar reasons to the average value feature. It varied quite significantly based on the motion being tested but stayed within a margin of error for each given motion.Pressure time integral (PTI) of each sensor. This is the summation of each pressure value multiplied by its corresponding sample value (Feature 13 to 18). The PTI was calculated using the following equation: $\sum_{t=1}^{N}P_{i}\left(t\right)\times \Delta t$ where *N* was the total number of samples in a data segment, *i* was the index of the sensors (1–6), *Pi* was the sensor value at sample number *t*, and Δ*t* is the number of samples from the beginning of the data segment [21]. The pressure time integral was a feature that helped differentiate between motions of different lengths. By summing the relative pressure values by the sample time, a greater variation between motions of various length was provided. This helped to increase the accuracy and allowed for more motions to be added into the study and classified accurately.

The data were then processed to a Tensorflow 2 machine learning algorithm, and several algorithms were tested to determine the highest accuracy algorithm. Of these algorithms, k-NN was selected due to the higher accuracy compared to the other algorithms. The k-NN algorithm is a supervised machine learning algorithm that operates on the assumption that similar things exist near one another. It works by finding the distance between points on a graph and chooses a value k and picks the first k entries that are closest to a certain point and captures their classification labels. The algorithm is trained by choosing different values for k and selecting the value which results in the most homogenous classification possible, while attempting to maintain the prediction accuracy as more unknown data were input. For our algorithm, a k value of one was chosen, as there was a lot of overlapping data and which would allow the algorithm to select more than one nearest neighbor, thereby resulting in significant misclassification. The algorithm also used a Euclidean distance metric to choose this neighbor with equal weighting given to distance. Dimensional reduction was applied to the data using PCA. The dimensions of the features were reduced to 17 from the original 18 features. This increased the separation between classes and decreased the training time. For the validation of the k-NN model, four-fold cross-validation was used. Each subject’s data was limited to one-fold, ensuring independence of the data. Four folds were chosen to split groups into seven subject groups, based on the movements with lowest amount of data points. Furthermore, Figure 3 shows the progression from raw data to classification for this study.

## 3. Results

This section describes data collected using P2S2 for movement classification. Six FSR sensor data are shown as a function of samples. These are representative of what a typical movement would look like for each movement. Thirty-four participant’s data were included for the study including 22 female and 12 male participants with an average age of 22.6 years. Data were collected using P2S2 for movement classification.

### 3.1. Walking

Walking was chosen, as it was the most common motion any person will perform and the one that current technology tracks best. Figure 4 showed the progression of the motion. The back inside and outside sensors peaked first, showing the heel strike and the subsequent contact of the front of the foot with the floor as the other four sensors peaked afterwards. Three steps were shown.

### 3.2. Running

Running was considered as a movement similar to walking, but at a faster movement rate. Steps are more rapid and a larger GFR is displayed. Figure 5 shows the motion as seen by the sensors. Compared to walking, the strike of the heel was much more instantaneous and leaned heavily towards the outside. This was then followed by an almost immediate contact of the front half of the foot with the ground. It could also be seen that the number of samples from beginning to the end of the motion was much less, showing the short amount of time in which the foot was on the ground. Three steps of running were shown.

### 3.3. Walking Up and Down the Stairs

Going up and down the stairs was considered to be a very common motion during a person’s average day, so this was also included in the selected motions. Figure 6 illustrates an example of walking down the stairs. shows the heel striking the ground first followed by the front of the foot. Compared to walking, the motion was similar, but the front of the foot was under more pressure than the rear of the foot. Three steps are presented in the figure. Figure 7 presents two steps during stair descent. It shows a brief heel strike followed by a significant strike of the front of the foot. 

### 3.4. Stooping

The stooping motion, or kneeling with one foot forward, was chosen as a repeatable motion that could be predicted using our methods. Figure 8 and Figure 9 show the sensor data for the left foot for stooping with both the left and right foot forward. One full stooping motion is shown. Figure 8 showed very low sensor readings for the first half of the motion, as the subject put all their weight on the right foot as they kneeled with their right foot forward. Then, as they began to rise again, all their weight was placed on the front inside of their left foot which can be seen by the large peak of the sensor located under the big toe. As the participant’s foot shifted to being level with the ground, a small peak was seen on the sensors under the heel. Figure 9 demonstrates a large peak of the sensors under the heel initially as the participant stooped with their left foot forward and put most of their weight on the back of their left foot. Very little pressure was put on anything but the rear of the left foot during this motion.

### 3.5. Squatting

Another motion was squatting, as shown in Figure 10. The data show that a squat entirely concerned with the heel of the foot. They show that the rear two sensors under near constant load, with rising pressure observed as the user pushed back up to standing and then as they balanced the load on the foot once standing. We also noticed when looking at the data that some subjects performed squats with their weight on the front of their foot and their heels entirely off the ground. This did not prove to be an issue for classification though, as our data labelling and supervised machine learning scheme proved to allow for two quite different motions to be accurately classified as the same motion.

### 3.6. Pushing and Pulling a Cart

Pushing and pulling are two other motions that were chosen for this study. Looking at the sensor data for pulling in Figure 11, there was first heel contact with the ground and then contact with the front of the foot at nearly the same amplitude and duration. This was a similar motion to walking, but from our results we can tell that they can be differentiated between. Figure 12 showed two steps of pulling the cart. The sensor data show the front of the foot contacting the ground first with pressure predominantly on the very front of the foot. They then showed a light contact of the back foot, indicating that the subject was nearly on their toes as they walked backwards.

### 3.7. Falling

One of the most critical motions to detect was falling. This had huge applications for geriatrics and in anyone with disabilities. Falling backwardss and forwards were quite simple to be detected, as falling backwards included only peaks on the back of the foot in Figure 13, while falling forward included a significant shifting of weight from the back to front of the foot in Figure 14. The best indicator for falling left or right was strong maximums on the outside or inside middle of the foot. As a person falls, almost all their weight transfers to the outside of one foot and the inside of the other. The left foot data in Figure 15 was selected for detecting falling to the left and the right foot data in Figure 16 for falling to the right. This difference was easily detectable. 

### 3.8. Machine Learning Results

The machine learning algorithm was trained on a labelled dataset that consisted of all of the movement results for each subject within the study. To achieve the highest classification accuracy possible, many different ML algorithms were trained. Table 4 presents various machine learning schemes and their respective overall accuracies is presented below. All methods were trained using four-fold cross-validation as specified earlier.

With the number of datapoints in the dataset, the algorithm took only a few seconds to train using a Tensorflow 2-based classification algorithm utilizing CUDA acceleration via a GTX 1060 graphics processing unit. This was a significant advantage of the k-NN algorithm as compared to more complex machine learning techniques such as deep learning, which can take anywhere from minutes to hours to train. This processing time did not include the time it took to pre-process the data. This process included sorting all the data, segmenting them into usable datapoints, and then extracting features from them. Below, displayed in Figure 17, is a confusion matrix showing the results of the predicted and actual movements trained on data from all subjects.

The movement detection result showed high accuracy in predicting all movements, with a greater than 83% accuracy acquired for twelve of the thirteen movements and an overall classification accuracy of 90.4%. Algorithm confusion occurred with similar movements. The algorithm confused different types of falling, with falling right being misclassified with falling forward. Overall, the classified movements were accurate.

## 4. Discussion

Of the thirteen movements examined in the study, twelve had an accuracy of 83% or higher. Some difficulties occurred with the prediction of falling in the right direction. The model still predicted that a fall occurred but was confused with forward and left directions. One reason of this misclassification could potentially be due to the fitment of shoes on participants. Not all human feet of the same size are the same, and slight differences in shape could cause slight classification error. Increasing the sample size of participants could mitigate this issue by training on more people from a group of participants.

Compared to our proposed P2S2, there are limitations within the design of the sensor system. First, the number of pressure sensors can have both advantages and disadvantages in design. Utilizing more sensors produces a higher resolution of pressure distribution and machine learning accuracy at the cost of increased design complexity and monetary cost. For example, the F-Scan 64 developed by Tekscan located in South Boston, MA, USA [50] is composed of 64 pressure sensors. With a sensor density of 3.9 (sensels/cm^2^), this system costs approximately USD 7000. Secondly, a larger scanning area could be achieved with a bigger FSR sensor [51]. However, the sensor area is limited to the size of the shoe. Lastly, the sensitivity of a sensor can be improved using an advanced material and technology such as a piezoelectric sensor [52], carbon nanotube [53], and capacitive materials [54].

The machine learning algorithm selected for this study was a kNN based algorithm. There are many other algorithms suited for this application such as convolutional neural network (CNN), SVM, decision tree, and linear regression. kNN is a relatively small model with a quick training time. SVM and decision tree are simple models that can differentiate various human movement. Linear regression can find correlation based on pressure data to identify human movement. CNN can be accurate with optimized hyper parameters with a number of neurons, number of layers, and epochs for movement classification 

The applications of this technology could dramatically improve quality of life by understanding daily movements. For example, employees can be properly trained by knowing their movements and correction if a movement abnormality is detected. This opens the ability to prevent injury, thereby saving both personal and corporate expenses. Additionally, by knowing the types and number of human movements, a calorie usage can be reported to humans. This provides the individual encouragement to exercise. By tracking previous to current movements, trends can be developed to indicate progression towards weight loss.The potential for every person to wear a shoe that can help train them to perform tasks with less strain to their bodies, as well as to inform them of what they can do to become the healthiest version of themselves could be revolutionary in a world where personal health is becoming more and more important. Future work could focus on the areas of microcontroller integration, automatic data processing, the ability to export data wirelessly, as well as wireless charging of the system. These future goals all serve the idea of creating a fully integrated system that has mass marketability and is as easy to use as possible.

## 5. Conclusions

In summary, a low-cost non-invasive footwear P2S2, using six force sensitive resistors with machine learning techniques, was presented to demonstrate the prediction of human movements. A total of 34 participants, with an average age of 22.9, were tested with P2S2 at the Capstone College of Nursing in the University of Alabama. Thirteen commonly used human movements including walking, stooping left and right, pulling a cart backward, squatting, descending, ascending stairs, running, and falling (front, back, right, left) were predicted using kNN machine learning algorithm. Validation of model was performed using a 4 k-fold process, which isolated training and test data. The results of this study showed that the proposed P2S2 can predict almost all the thirteen different human movements with an average accuracy of above 86%, while falling right was classified at a 78% accuracy.

## Figures and Tables

**Figure 1 sensors-22-02743-f001:**
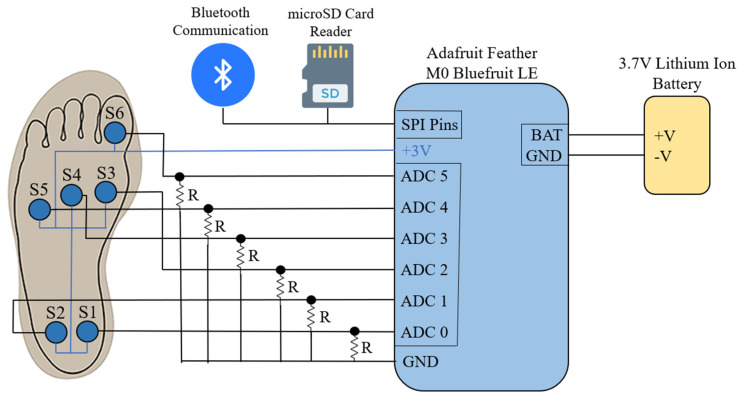
Schematic of the proposed pressure sensing system.

**Figure 2 sensors-22-02743-f002:**
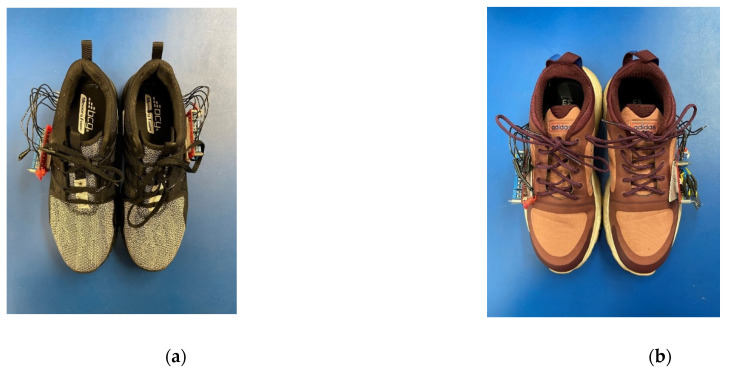
(**a**) men’s size 10.5 shoe with integrated insole pressure system and microcontroller shown. (**b**) women’s size 8.5 shoe with insole pressure system and microcontroller shown.

**Figure 3 sensors-22-02743-f003:**
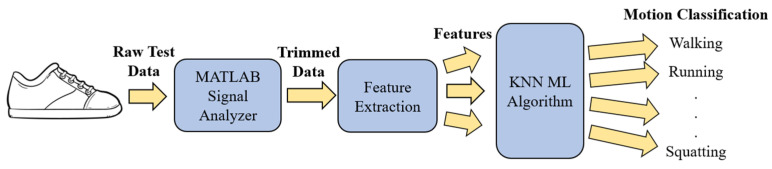
Diagram showing progression from raw data to classified movement.

**Figure 4 sensors-22-02743-f004:**
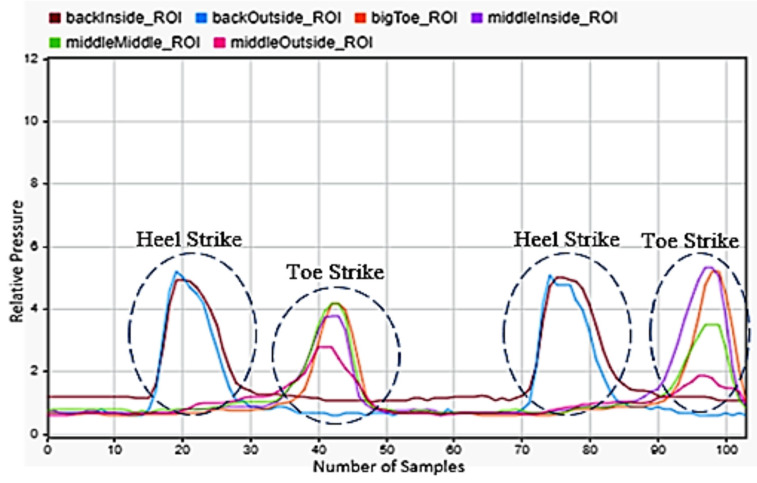
Walking motion sensor readings for the left foot.

**Figure 5 sensors-22-02743-f005:**
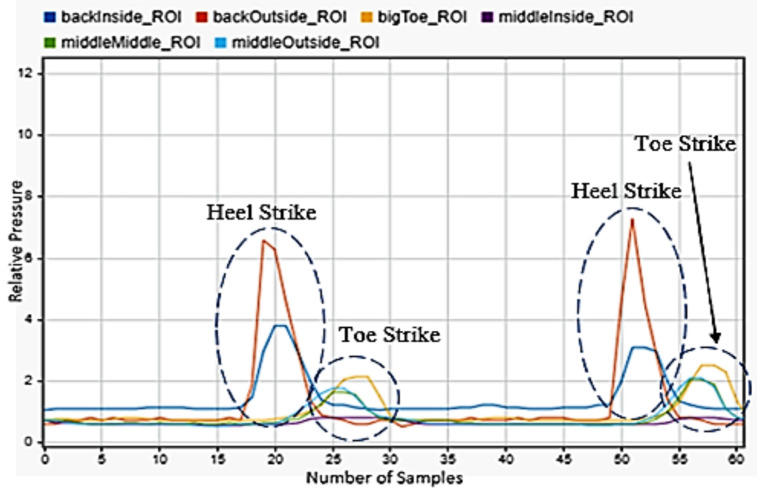
Running motion sensor readings for the left foot.

**Figure 6 sensors-22-02743-f006:**
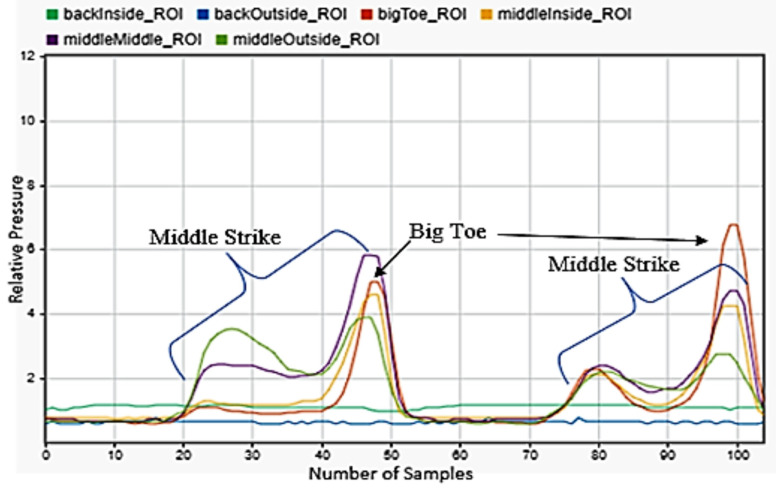
Stair ascent sensor reading for the left foot.

**Figure 7 sensors-22-02743-f007:**
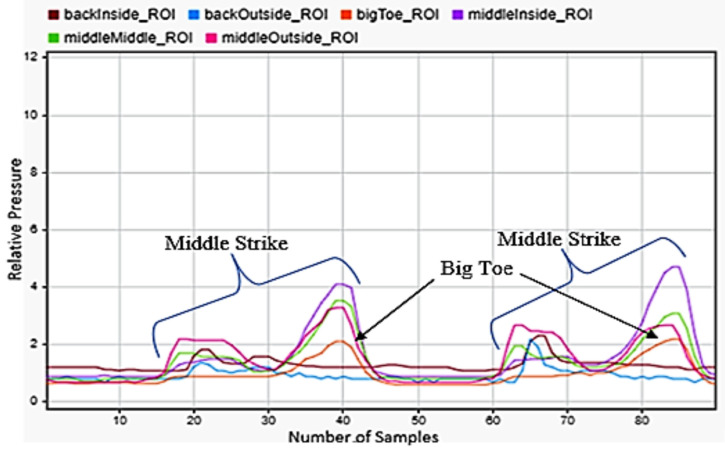
Stair descent sensor readings for left foot.

**Figure 8 sensors-22-02743-f008:**
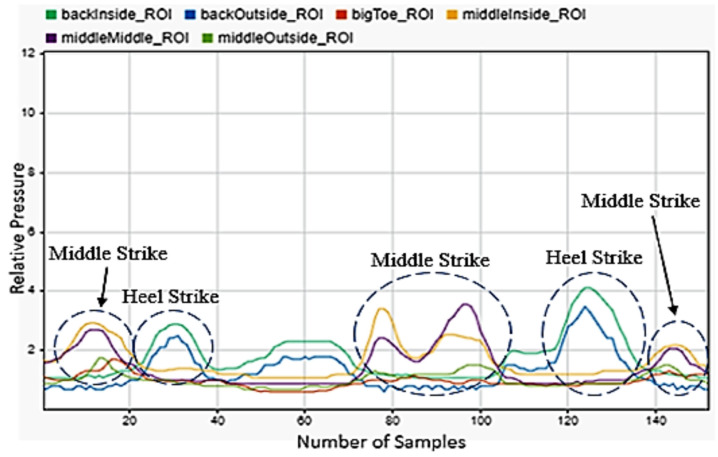
Stoop with right foot forward sensor readings for the left foot.

**Figure 9 sensors-22-02743-f009:**
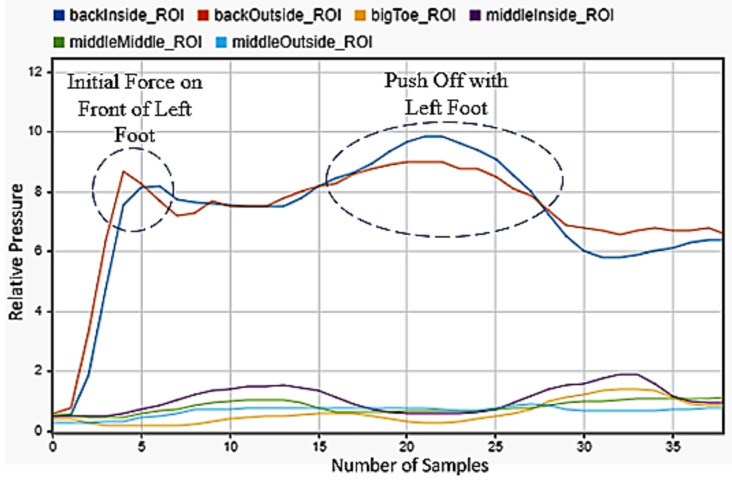
Stoop with left foot forward sensor readings for the left foot.

**Figure 10 sensors-22-02743-f010:**
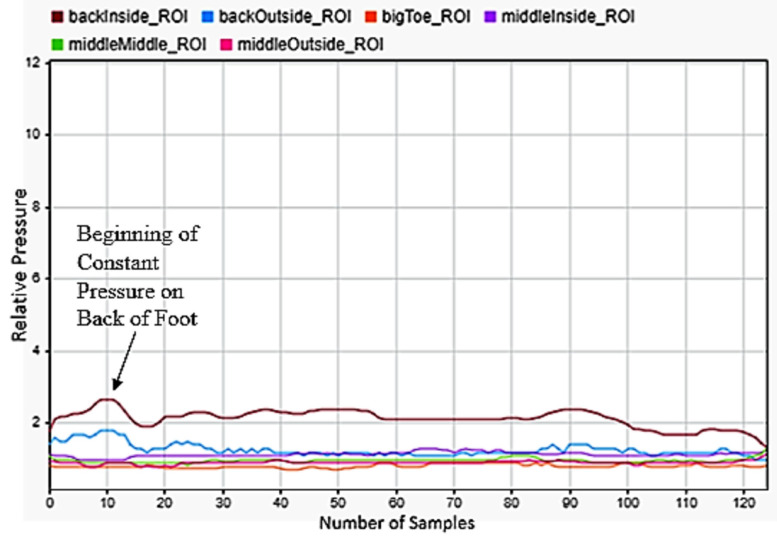
Squatting motion sensor data for the left foot.

**Figure 11 sensors-22-02743-f011:**
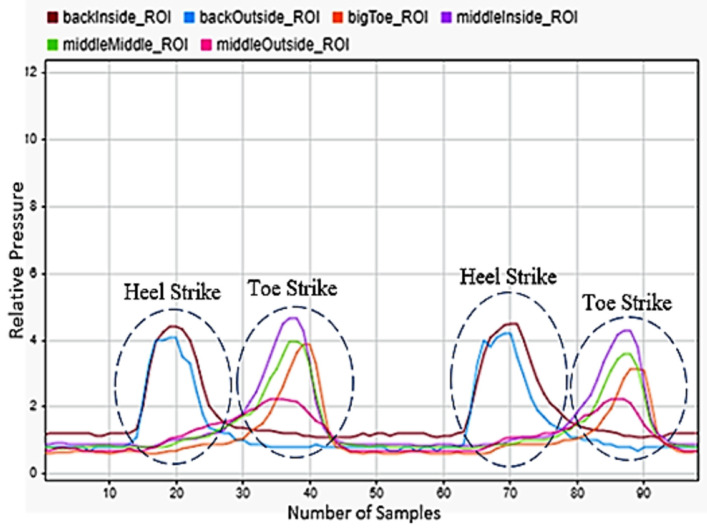
Pushing a cart sensor data for the left foot.

**Figure 12 sensors-22-02743-f012:**
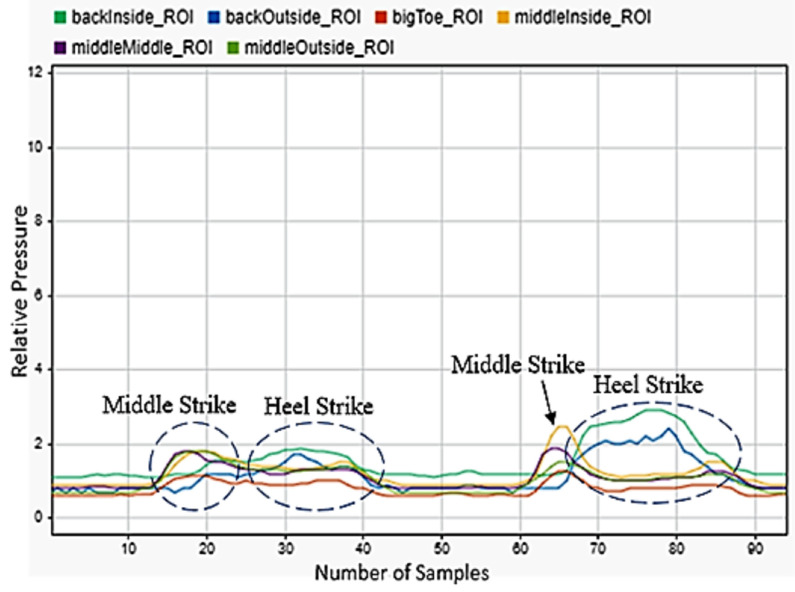
Pulling a cart sensor data for the left foot.

**Figure 13 sensors-22-02743-f013:**
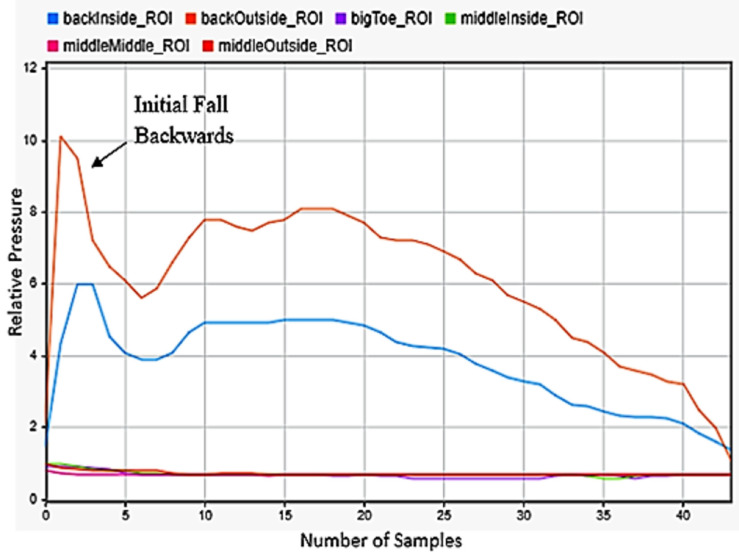
Falling backwards sensor data for the left foot.

**Figure 14 sensors-22-02743-f014:**
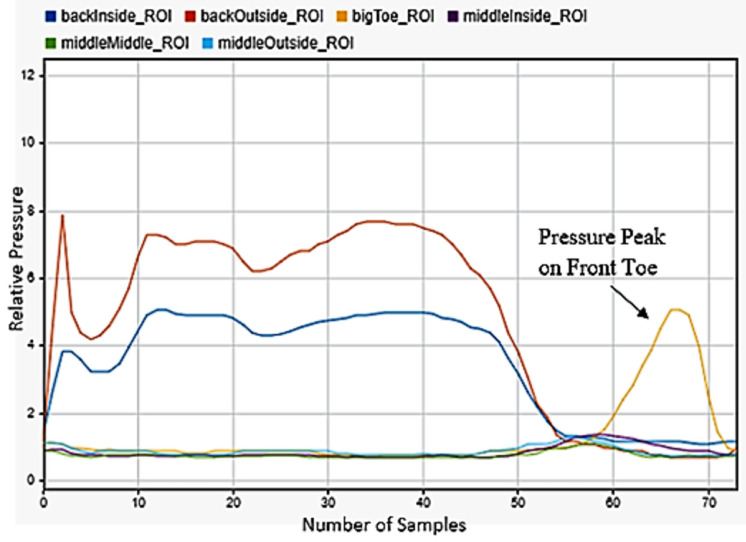
Falling forwards sensor data for the left foot.

**Figure 15 sensors-22-02743-f015:**
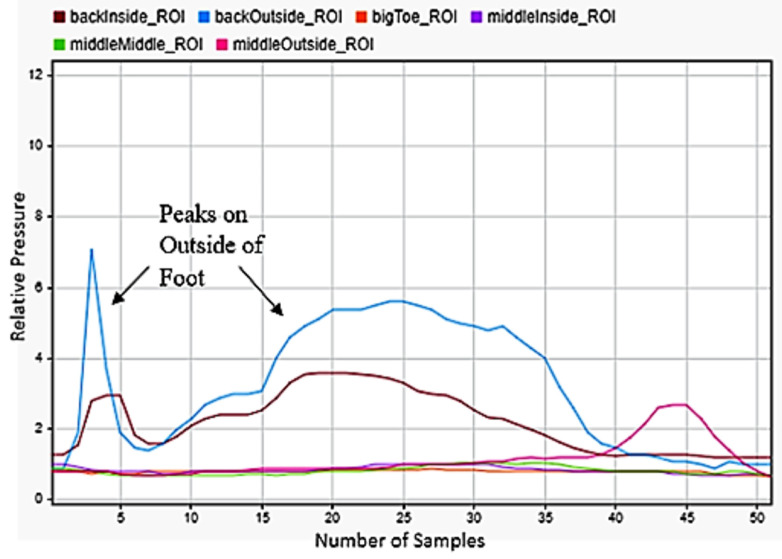
Falling to the left sensor data for the left.

**Figure 16 sensors-22-02743-f016:**
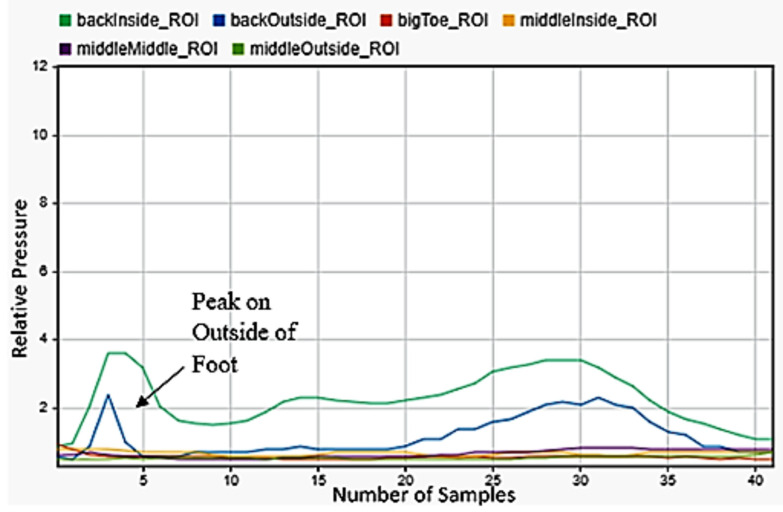
Falling to the right sensor data for the right foot.

**Figure 17 sensors-22-02743-f017:**
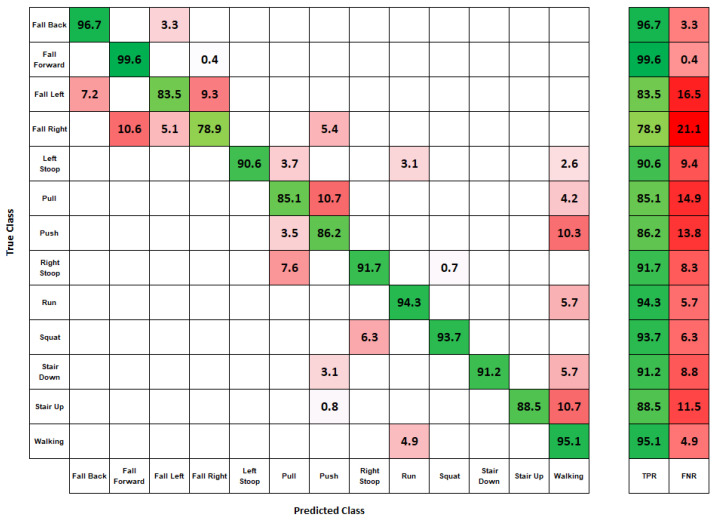
Movement detection results from 34 subject data. True Positive Rate (TPR) and False Negative rate (FNR).

**Table 1 sensors-22-02743-t001:** Previous Studies for Human Movement Detections.

Author	Application	Sensor	Number of Sensors Used	Machine-Learning Algorithm	Type of Movements	Accuracy
Crema et al. [5]	Physical movement	IMU	1	Linear Discriminant Analysis, Principal Component Analysis	9 gym exercises (bench press, squats, shoulder press, etc.)	85%
Lu et al. [6]	Physical movement	IMU, Image	5	Capsule Networks, Convolutional Long short-term memory (LSTM)	6 cooking activities (opening fridge, cracking eggs, stirring eggs, pouring oil, pouring bag, stirring big bowl)	85.8%
Lao et al. [10]	Physical movement	Video	1	Continuous Hidden Markov Model	Left/right hand pointing, squatting, raising hands overhead, lying	86%
Geng et al. [12]	Physical movement	On-body radio freqency (RF) receivers and transmitters	5	SVM	Standing, walking, running, lying, crawling, climbing, and running up stairs	88.69%
Wang et al. [13]	Physical movement	Acoustic	2	None	Respiration	None
Yun et al. [14]	Physical movement	Infrared	4	Bayes Net, Decision Tree, Instance-based learning, Multilayer Perception, Naïve Bayes, SVM	Walking in different directions	99.9%
Hegde et al. [23]	Physical Movement	FSR, accelerometer, and IMU	13	Multinomial logistic discrimination	Lying, sitting, standing, walking, driving, stair descent/ascent, cycling, vacuuming, shelving items, dish washing, sweeping, not wearing device	89%
Jeong et al. [24]	Physical movement	FSR	3	SVM	Walking, stair ascent/descent	95.2%
Antwi-Afari et al. [26]	Physical movement	Capacitive	4	SVM	Lifting, lowering, carrying, standing	94.4%
Sazonov et al. [27]	Physical movement	Accelerometer and FSR	6	SVM	Sitting, standing, walking, stair ascent/descent, and cycling	98%
Nguyen et al. [29]	Physical movement	FSR	5	SVM	Walking on flat, inclined, or declined surface, stair ascent/descent	97.8%
Leu et al. [30]	Physical movement	Mobile phone	2	Decision tree	Six types of falls	96.57%

**Table 2 sensors-22-02743-t002:** Detailed description of all recorded motions.

Motion Name	Figure	Description	Duration (Minutes)
Falling (split into 4 directions: left, right, forward, backward)	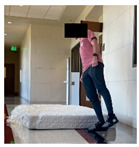	Subject fell a total of eight times on to a full-sized mattress of approximately 7-inch thickness. The falls occurred two times in each of the following directions: forward, backward, on the right side, and left side.	3
Stoop Left	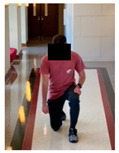	Subject performed a kneeling motion with their left foot forward and then stood back up. This was repeated until ten stoops had been completed.	2
Pulling a cart backward	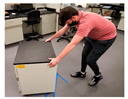	Subject walked backwards and pulled the cart five steps. This was repeated once for more data points.	1
Pushing a cart forward	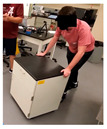	Subject pushed the cart approximately five steps forward. This was repeated once for more data points.	1
Stoop Right	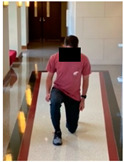	Subject performed a kneeling motion with their right forward and then stood back up. This was repeated until ten stoops had been completed.	2
Squatting	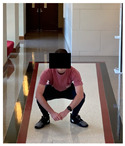	Subject stood still, squatted down, and then returned to a standing position. This was repeated ten times.	2
Descending stairs	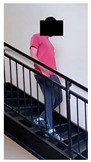	Subject naturally descended stairs. The stairs will be a standard flight located at the test site.	1
Ascending stairs	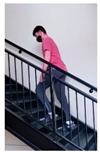	Subject naturally ascended stairs. The stairs will be a standard flight located at the test site.	1
Running	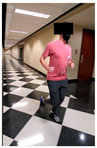	Subject jogged down a hallway at the test site (approximately 30 steps), repeating once for more data points.	1
Walking	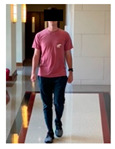	Subject walked down the same hallway at the test site This was then repeated once more for more data points.	2

**Table 3 sensors-22-02743-t003:** Participant’s information.

Subject	Sex	Age	Height (Inch)	Weight (Lb)	Shoe Size (Inch)
1	Female	21	5′3″	120	8.5
2	Female	21	5′4″	185	8.5
3	Female	21	5′7″	130	10.5
4	Female	21	5′7″	135	8.5
5	Female	41	5′1″	150	8.5
6	Male	21	5′11″	180	10.5
7	Female	21	5′9″	170	10.5
8	Female	21	5′8″	125	8.5
9	Female	21	5′4″	165	8.5
10	Male	21	6′1″	170	10.5
11	Female	20	5′7″	140	8.5
12	Male	24	5′10″	185	10.5
13	Male	21	5′11″	170	10.5
14	Female	20	5′7″	140	8.5
15	Female	29	5′3″	145	8.5
16	Male	23	5′10″	175	10.5
17	Male	21	6′1″	150	10.5
18	Female	21	5′4″	150	8.5
19	Female	23	5′5″	155	8.5
20	Male	19	6′1″	135	10.5
21	Male	21	5′8″	160	10.5
22	Male	44	6′0″	205	10.5
23	Female	22	5′8″	150	8.5
24	Male	22	5′11″	145	10.5
25	Female	22	5′4″	165	8.5
26	Female	21	5′10″	135	8.5
27	Female	20	5′6″	130	8.5
28	Female	21	5′6″	138	8.5
29	Female	20	5′6″	145	8.5
30	Female	21	5′5″	140	8.5
31	Male	22	6′3″	190	10.5
32	Female	20	5′2″	112	8.5
33	Female	21	5′6″	140	8.5
34	Male	21	6′0″	145	10.5

**Table 4 sensors-22-02743-t004:** Machine Learning Algorithm Performance Comparison.

ML Algorithm	Details	Epochs	Training Time	Accuracy
SVM	Quadratic kernel function, 1-vs.-1 multiclass method	1000	25.1 s	89.9%
Neural Network	Medium NN, one fully connected layer, first layer size of 25	1000	27.1 s	89.2%
Neural Network	Wide NN, one fully connected layer, first layer size of 100	1000	34 s	89.5%
KNN	Weighted, 10 neighbors, Euclidean distance metric, squared inverse distance weight	1000	25.1 s	90.4%

## Data Availability

Not applicable.

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
