# Peer review of "Empirical Study on Human Movement Classification Using Insole Footwear Sensor System and Machine Learning"

_sensors, 2022, doi:10.3390/s22072743_

Round 1

Reviewer 1 Report

GENERAL COMMENT

This paper presents the development of a pressure system and classification algorithm to enable the identification/classification of several different common activities (including, but not limited to, walking, running and stair ascending). This is certainly a topic of interest today, as the technological advancements (e.g., miniaturization of sensors, long lasting batteries) and the availability of wearables to collect data in the real world (outside of the lab) are leading to the widespread use of wearables to monitor health status and mobility. To this end, the development of reliable classification systems is crucial.

While the scope of the paper is clear, the Methods section could be improved to provide the readership with more (key) details to replicate the results and/or better understand how the proposed system was defined and trained. The Results section should be restructured (to improve clarity) and a Limitations section should be added.

In its current form, the manuscript requires major revisions before it can be accepted for publication, as detailed in the comments below.

SPECIFIC COMMENTS

Introduction

  1. Line 18: […] processed using PCA (Principal Component Analysis)

It should read Principal Component Analysis (PCA). Please correct.

  1. Line 20: SVM is not defined

Please define the acronym at first use.

  1. Line 29-30: […], the IMU revealed discomfort in using daily life because of many IMU that needs to be attached to a body […]

It should read ‘daily life use/using’. Please correct.

More importantly, please provide references supporting this statement. In fact, one IMU sensor may suffice, with appropriate algorithms to process the recorded data.

  1. Line 41: […] relatively accurate as well […]

Please provide reference(s).

  1. Line 45-46: Another solution to this problem is to use a smart shoe using pressure sensing technology in combination with machine learning.

Please list the relevant references.

  1. Line 46-47: The main area of investigation in this area […]

Please check wording to avoid repetitions and improve clarity.

  1. Line 54: […] and keep aside the sensors was classified.

Please check wording to improve clarity.

  1. Line 57: Many studies also showed detection of falling [21,22].

Please add some references or change the word ‘many’.

  1. Lines 58-60: Based on previous research, there is a gap in knowledge related to more diverse movements, and a system which can detect a broader set of human movements is in high demand for development.

Ref 23 already reports the classification of 13 activities. Could the authors please specify the rationale for the need to enable the classification of more than 13 daily activities?

  1. Please consider adding a table summarizing the various available methods to capture human movement, with their level of accuracy and references.

  1. Table 1

Please define all acronyms in the table caption.

  1. Lines 64&66

Please add complete manufacturer’s details (Company name, Country, City).

Please add a reference for the pedar system, if available.

  1. Lines 71-74

Please reword to clearly state (1) what the aim was and (2) the work that was done to achieve it. The footwear system was designed and described in previous papers (refs [27],[46]). The focus here appears to be on the classification algorithm.

Methods

  1. Line 86-87: […] this force will vary and will be at different points on the foot.

Please consider rephrasing for sake of clarity (e.g., the GRF will vary in magnitude and location (point of application on the foot)).

  1. Line 119

Please also list in the manuscript main text the fourteen tested activities.

  1. Line 121: […] reliably repeatable for testing.

Please provide references to support this statement and/or indicate which of the fourteen activities belong to this category.

  1. Line 122: […] in the following table.

Please refer to tables and figures using their numbers. In this case would be Table 2.

  1. Table 2 (Running): […] ., repeating […]

Please check spelling.

  1. Line 128: 50Hz

Please provide the rationale for this choice and supporting references. Is it valid for tasks such as running?

  1. Lines 132-133: Individual movement tests were separated from the text file using a MATLAB script.

Please provide details on how this was performed.

  1. Please consider inverting section 2.3 and 2.4, so that there is a chronological order (from experimental measurement to data processing and classification).

  1. Line 147: […] participants were asked to perform a heel raise before a new motion was performed.

Please explain the rationale for this choice (why the heel raise?). I suppose it is because the GRF profiles during heel raise are peculiar and easily distinguishable from those recorded during the other tested tasks.

  1. Line 153: The figure below […]

Please correct, as it is a table, not a figure. Furthermore, please use table numbering.

  1. Table 3, headings

Please add units.

  1. Lines 161-163

Please provide details on the informative value of each feature, to explain why these were chosen (over others).

  1. Lines 171-172

Please check wording. As it is, it does not make full sense.

  1. Line 183: Principal component analysis

PCA was already defined above.

  1. Please provide a rationale for the choice of four as k in the k-fold cross-validation.

Results

  1. Please consider restructuring the Results section. Since the focus of the manuscript is primarily on the classification algorithm (as the pressure sensor was described and validated in previous work), please reduce the description of the tested movements and the profiles recorded by the sensors (Sections 3.1-3.7) – perhaps highlighting how the recorded profiles are as expected, enabling to identify specific features. Consider moving some of the Figures to Supplementary material and include only few in the main text.
  2. Please clarify whether the profiles showed in all these Figures are representative of one specific subject or whether they can be considered representative of the whole tested population.
  3. Please consider adding vertical lines to the figures to mark specific events (e.g., heel-strike and toe-off for walking) or key features.
  4. Please add units of measurement to the axes of the Figures.
  5. Lines 201-202

              Please reword the first sentence to improve clarity

  1. Please consider adding a table to recap the overall time expenditure (for each block of the proposed pipeline). It could be important for the readership to know.
  2. Figure 16

Please define the acronyms in the caption.

  1. Please add reference threshold values for the accuracy to be considered acceptable and contextualize (as the acceptability level changes based on the final goal).

Discussion

  1. Line 316: Of the thirteen movements examined […]

Please check the number. Sometimes it is mentioned that 14 different human movements were acquired during testing.

  1. Please add a limitation section (e.g., did the feet perfectly fit in the shoes, for all participants? 10.5 and 8.5 sizes are average sizes for men and women in the US, but there may be slight differences from person to person that could affect the pressure measurements).

Conclusion

  1. Please consider moving this section to last
  2. Please mention which movement (of the fourteen tested) was not accurately classified.

Round 2

Reviewer 2 Report

I appreciate your efforts to make the changes in the article, but discussion section is still vague. It is extremely short. Lack of reasoning. It is necessary to compare it with other similar research studies, as well as to add limitations of the study.

Author Response

Thank you very much for your valuable comments.
